# Rapid topographic growth of the Diancang Shan, southeastern margin of the Tibetan Plateau since 5.0–3.5 Ma

Chunxia Zhang[1,2], Haibin Wu[1,3], Xiuli Zhao[2], Yunkai Deng[1,3], Yunxia Jia[4], Wenchao Zhang[1], Shihu Li[5], Chenglong Deng[3,5]

[1]Key Laboratory of Cenozoic Geology and Environment, Institute of Geology and Geophysics, Chinese Academy of Sciences, Beijing 100029, China
[2]College of Earth Science and Engineering, Shandong University of Science and Technology, Qingdao 266590, Shandong Province, China
[3]University of Chinese Academy of Sciences, Beijing, China
[4]School of Geographical Science, Shanxi Normal University, Taiyuan 030031, China
[5]State Key Laboratory of Lithospheric Evolution, Institute of Geology and Geophysics, Chinese Academy of Sciences, Beijing 100029, China

*Correspondence to*: Chunxia Zhang (cxzhang@mail.iggcas.ac.cn)

**Abstract.** As a crucial geological, climatic, and ecological boundary in the southeastern margin of the Tibetan Plateau (SEMTP), the topographic evolution of the Diancang Shan (DCS) remains unclear due to the lack of direct constraints on its paleoelevation. Here, we quantitatively reconstructed changes in annual mean temperature (*ANNT*) based on palynological data from the terrestrial Dasongping section (~7.6–1.8 Ma) in the Dali Basin, located at the northeastern margin of the DCS in Yunnan Province, China. Integrating the thermochronological data from the eastern and southern margins of DCS, we have clarified the paleotopographic evolution of DCS during this period: the paleoelevation of DCS likely exceeded 2000 meters above sea level (m a.s.l.) due to initial normal faulting at ~7.6 Ma, possibly comparable to the current average elevation (~2200 m a.s.l.) of surrounding Dali Basin region. Significant growth occurred between ~5.0 Ma and ~3.5 Ma, with at least ~1000 meters uplift gain in the northern segment and up to ~2000 meters in the southern segment of DCS, caused by the intensification of normal faulting activities. Finally, the northern segment of DCS reached the elevation of ~3500 m a.s.l. after ~1.8 Ma. Our findings suggest that the quantitively *ANNT* reconstruction, combined with thermochronological and sedimentary data, can significantly improve constraint on the paleotopographic evolution of DCS.

## 1 Introduction

The southeastern margin of Tibetan Plateau (SEMTP) has evolved tectonically as an important accommodation zone during the post-collisional intracontinental deformation between Indian and Eurasian continents (Tapponnier et al., 1990; Houseman and England, 1993; Leloup et al., 1995; Clark and Royden, 2000; Clark et al., 2004; Clark et al., 2005). Much of the intracontinental deformation of SEMTP during Cenozoic has been accommodated by strike-slip fault systems, of which the Ailao Shan-Red River shear zone (ASRRSZ) (Fig.1a) is the largest one (Harrison et al., 1995; Leloup et al., 1993; 1995; Tapponnier et al., 1990; Wang et al., 1998; 2006). The last transformation of shear zone with dextral ductile-to-brittle

transitional normal faulting in late Cenozoic was thought to have resulted in large-scale uplift and erosion, leading to the formation of today's mountains within the ASRRSZ (Allen et al., 1984; Tapponnier et al., 1990; Leloup et al., 1993; 1995; Wang et al., 1998; 2006). Diancang Shan (DCS) lying in the northwestern part of the ASRRSZ in SEMTP (Fig.1a), is a transition belt of the high Tibetan Plateau to the low relief of East and South Asia and a neighbouring belt of subtropics to the east and middle subtropics to the west. As a significant geological, climatic, and ecological boundary within the SEMTP, the tectonic evolution of DCS provide great information on our understanding of the uplift process of Tibetan Plateau (Tapponnier et al., 1990; Leloup et al., 1993; Harrison et al., 1995; 1995; Wang et al., 1998; 2006; Wang et al., 2020a; 2020b; 2022).

Clarifying the paleoelevation changes of DCS is important for comprehending both the tectonics and the climatic effects in this unique geographical location. Most of thermochronological studies primarily focused on tectonic activities surrounding DCS range, revealing that the extensive right-lateral strike-slip movement in this region experienced numerous episodes of obvious shear displacement at ~8.0 Ma, ~5.0 Ma, 2.7 Ma, and 2.1 Ma, respectively (Leloup et al., 1993; Harrison et al., 1995; Fan et al., 2006; Xiang et al., 2007; Cao et al., 2011; Han et al., 2011; Li et al., 2012). Only very few thermochronological studies mentioned that DCS had reached its maximum elevation at ~4.7 Ma, followed by significant unroofing during Quaternary (Leloup et al., 1993). However, the existence of relict surface on the top of DCS indicates that there was no unroofing during the late Cenozoic. Wang et al. (2006) suggested that the rock cooling events around DCS revealed by low-temperature thermochronological data were not caused by unroofing but possibly resulted from normal faulting along mountain front. Schildgen et al. (2018) proposed that regional variations in exhumation rates can be accurately assessed by integrating local findings that encompass thermochronological data and location-specific information. Although numerous geochronological and thermochronological data exist, documenting multiple episodes of tectonic activities around DCS range, there is currently a significant lack of precise constraints on the paleoelevation of DCS throughout the period since the late Cenozoic.

The coupling between sedimentary basins and orogenic belts is an important geomorphic evolutionary process. The Sanying Formation is widely distributed along piedmont and hill-front areas and in fault-controlled basins in Yunnan Province, Southwest China and has been used to constrain the reactivation of the sinistral Dali fault system and the timing of surface uplift of the SEMTP (Wang et al., 1998; Clark et al., 2005). The Sanying Formation spans the interval from the Late Miocene to Early Pleistocene (~7.6–1.8 Ma) in the Dali Basin, which is the most complete fluviolacustrine sequence in the NE flank of the Diancang Shan (DCS) (Li et al., 2013). The activity of the Dali fault system and the surface uplift of the DCS during the late Neogene have been reconstructed and constrained using chemical compositions and bulk mineralogical characteristics of sedimentary rocks from the Dasongping section (Zhang et al., 2020). As mountain elevation has a direct impact on climate and vegetation, paleobotany recorded in sedimentary basin can be utilized as a proxy for estimating paleoelevation. Pollen assemblages in sedimentary sequences have been used to reconstruct paleoelevation in the Tibetan Plateau (Song et al., 2010; Sun et al., 2014) and indicate rise of the northeastern parts of Tibetan Plateau (Dupont-Nivet et al., 2008; Miao et al., 2022). In this study, we investigate palynological data of the late Neogene sedimentary record from the Dasongping section and quantitively reconstruct changes in the annual mean temperature (*ANNT*) based on pollen data. By combing *ANNT* with

sediment accumulation rate (SAR) of the Dasongping section and thermochronological data from the eastern and southern margins of DCS, we aim to quantitively reconstruct the paleoelevation evolution of DCS during late Cenozoic.

## 2 Materials and Methods

### 2.1 Geological setting and sampling

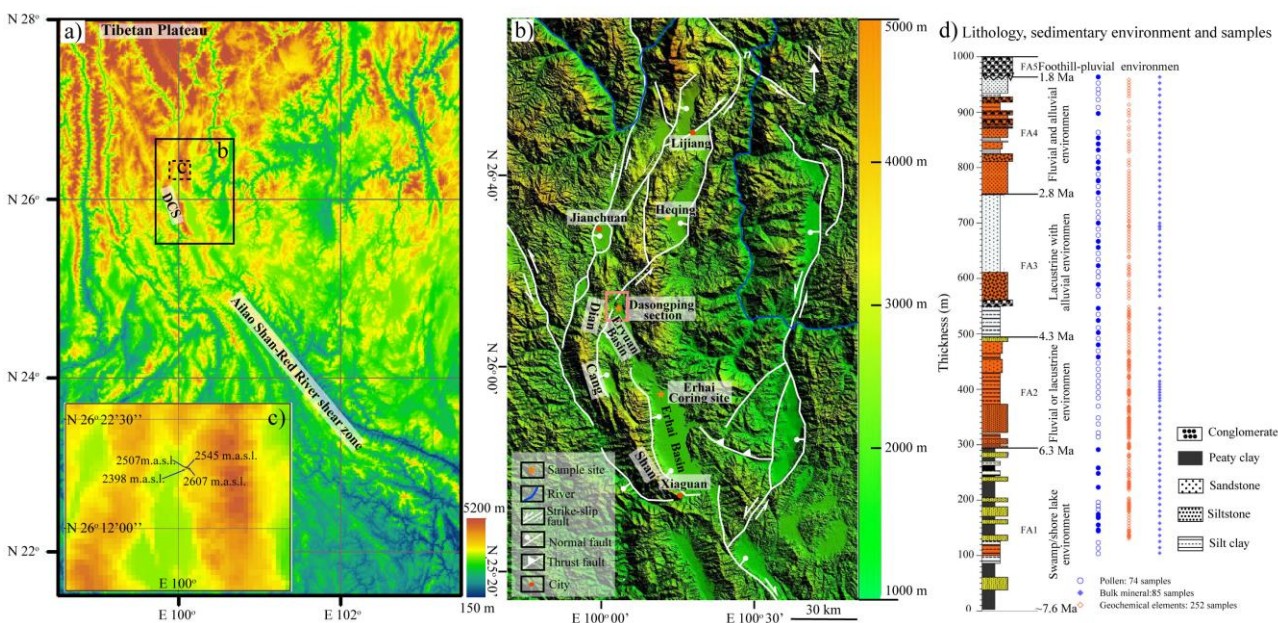

**Figure 1. (a) Location of the study area. (b) Shaded relief map of the Dali Highlands and the Dali fault system, as well as the Diancang Shan. The distribution and configuration of active faults are modified after Wang et al. (1998). (c) The altitude distribution of the four surface samples. (d) Lithologies, sedimentary facies, and samples for pollen, bulk mineral, and geochemical analyses. DCS:**
**Diancang Shan. Solid blue circles in (d) correspond to samples that yielded abundant pollen grains.**

The present landscape of DCS is an elongated metamorphic massif, with a length of 80 km and width of ~50 km. It has an average elevation of 4000 m a.s.l. in the south segment, 3700 m a.s.l. in the middle, and 3500 m a.s.l. in the north segments, with the highest peak of 4122 m a.s.l. The top of the DCS is a very flat relict surface which can be traced from the northern
edge to the southern edge of the massif for 100 km (Wang et al., 1998; Fan et al., 2006). DCS is ~1800 m above the surrounding regions (average elevation ~2200 m a.s.l.), and exhibits vertical climatic zoning vegetation (Ming and Fang, 1982).

The eastern flank of DCS exhibits triangular shaped facets separated by deep canyons (Wang et al., 2006), suggesting its normal faulting activity. The Dali Basin formed in response to movement on the east-dipping Erhai normal fault system, with an elevation of 1980 m a.s.l., which is a pull-apart basin located to the east of DCS (Fig. 1a). It includes the Eryuan Basin in
the north and Erhai Basin in the south (Fig. 1b). Total thickness of Neogene-Quaternary sediments in the Dali Basin is about 2200 m (Guo et al., 1984). These phenomena suggest that the eastern flank of DCS corresponds to an active east-dipping

normal fault and the thick sediments in the basin result from rapid erosion of uplifted DCS (Guo et al., 1984; Tapponnier et al., 1990; Leloup et al., 1993). The Dali Basin has a northern tropical monsoon climate, with a mean annual precipitation of 750 mm and mean annual temperature of 13 °C (Chen, 2001).

Dasongping section preserves the best exposures of Neogene coal-bearing sedimentary rocks of the Sanying Formation (Li et al., 2013) in Eryuan Basin on the northern flank of DCS (Fig. 1b and d) at an elevations of 2174–2633 m a.s.l. The terrestrial record has a thickness of ~1000 m (Fig. 1d) and spans the interval from ~7.6 Ma to ~1.8 Ma, and is disconformably overlapped by unconsolidated Quaternary conglomerates and is underlain by Silurian limestone (Li et al., 2013). Lithological variations allow five facies associations (FAs) to be identified from bottom to top: swamp/shore lake facies (FA1, 0–300 m),
fluvial or lacustrine facies (FA2, 300–505 m), lacustrine with alluvial facies (FA3, 505–750 m), fluvial and alluvial facies (FA4, 750–965 m), and foothill–pluvial facies (FA5, 965–1000 m) (Fig. 1d) (Li et al., 2013; Zhang et al., 2020). Bulk samples were collected from the mudstones, siltstones, and sandstones of the Dasongping section at a ~1 m interval. A total of 74 samples with an average sampling interval of ~10 m was analyzed for their palynological assemblages. Additionally, 4 surface samples within elevations of ~2400–2600 m from the Dasongping section area (Fig.1c) were collected for their palynological analysis.

**2.2 Methods**

### 2.2.1 Pollen data and climate data

78 samples analyzed for pollen were treated with HCl (15%) and NaOH (~3%) to remove carbonate and organic matter, respectively. Pollen in each sample was concentrated using heavy-liquid separation with $ZnI_2$. Before mounting the palynomorphs in glycerin jelly, HF (40%) and HCl (36%–38%) were used to remove silicates. Finally, 31 out of the 74 samples
and 4 surface samples with average elevation of 2500 m.a.s.l. (Fig.1c) yielded abundant pollen grains, while very few pollen grains (<5 grains) or none were obtained from sandstone, silty sandstone and red siltstone (Fig. 1d, Sup Table 1). Excluding aquatic taxa, Pteridophytes and Cyperaceae, a minimum of 270 grains was counted per sample, and more than 300 pollen grains were counted for 25 samples (Fig.2, Sup Table 1). We calculated the pollen and spore percentages of each taxon base relative to the sums of terrestrial pollen grains and total grains, respectively. Palynological datasets of the late Holecene, mid–
Holocene from the Lake Erhai coring (Fig.1b) were compiled from the literatures Shen et al. (2006), and all pollen samples preparation methods were following Liu et al. (1986).

We use the modern surface pollen assemblage data of China compiled by Sun et al. (2020). This dataset comprises 1630 sample sites, covering most of the regions of China. The modern climate data from the WorldClim 2.1 database (Fick and Hijmans, 2017) were obtained. The database provided gridded climate data at a spatial resolution of 30 second (~1 $km^2$) and
was the average for the years 1970–2000.

### 2.2.2 Climate reconstruction

To reconstruct the history of vegetation evolution, we translated pollen assemblages into vegetation types using the biomization

procedure proposed by Prentice et al. (1996). This method classifies individual pollen taxa into taxa-plant functional types (PFTs) according to plant stature, leaf form, phenology, and climatic adaptations, and then into vegetation types (biomes) defined by the dominant PFTs. In this study, we calculated biome scores and reconstructed vegetation changes using the latest PFTs-biomes matrixes defined for the China region (Sun et al., 2020).

We used the PFTs-based modern analogue technique (MAT) method to quantitively reconstruct the annual mean temperature (*ANNT*) following the procedure proposed by Zhang et al. (2022a). Briefly, the original taxa were converted into PFTs scores. The modern analogues were identified using the squared-chord distance metric of dissimilarity. The dissimilarity-weighted climate of the 6 closest modern analogues was assigned to the fossil sample. The performance of this model was evaluated using the bootstrap cross-validation. The performance of weighted-average partial least squares (WAPLS) model was also evaluated with the same calibration set for comparison. All the analyses were carried out using R software (version 4.3.3; R Core Team, 2024). The predicted *ANNT* based on PFTs-based MAT showed more significant linear relationship ($R^2$=0.83) with observed *ANNT* than weighted-average partial least squares (WAPLS, component=3) ($R^2$=0.80) estimated by bootstrapping cross-validation (Fig.S1). It is reasonable that the scatter plots of PFT-based method exhibit higher dispersion than those of the WAPLS method, because the modern analogues might be impacted by non-climatic factors (Zhang et al., 2022b)

### 2.2.3 Paleoelevation reconstruction

Pollen assemblages in the sedimentary section were results of mixing at different elevations, the paleoclimate quantitive reconstruction based on the PFTs-based MAT method in the Dali Basin would be more sensitive to the variation of elevation. After excluding the influence of global temperature changes, the quantitively differences in *ANNT* values between surface samples at 2500 m.a.s.l. and sedimentary records record the topographic variations of DCS. To reduce the systematic error in estimating ancient elevations from pollen data, we employed the temperature mean anomaly, specifically quantifying the disparities in *ANNT* values between sedimentary archives and contemporary surface samples situated at 2500 m.a.s.l., as a means to reconstruct the palaeoelevation of DCS. The paleoelevation of DCS in per Ma interval was calculated, using the common temperature lapse rate (TLR) of 5–6 °C km$^{-1}$(Miao et al., 2022) , based on the following formula:

$$\text{Elev}_{paleo}=2500-1000*(ANNT_{\text{mean-anom}}-SST_{\text{mean-anom}})/\text{TLR} \quad (1)$$

$$ANNT_{\text{mean-anom}}=ANNT_{\text{mean}}-ANNT_{\text{mean@2500 m a.s.l.}} \quad (2)$$

$ANNT_{\text{mean}}$: the mean *ANNT* per Ma period of the Dasongping section, $SST_{\text{mean-anom}}$: tropical sea surface temperatude mean anomaly per Ma period. $SST_{\text{mean-anom}}$ data from the literatures Herbert et al. (2016) and Holbourn et al. (2018)); $ANNT_{\text{mean-anom}}$ and $SST_{\text{mean-anom}}$ are shown in Fig.6f.

### 2.2.4 Sediment accumulation rate and thermochronological data

Sediment accumulation rate (SAR) is an important proxy to reflect the intensity of sediment deposition in a basin, which is affected by sediment transport capacity, sediment source and climate etc. Therefore, SAR is of great significance for

understanding basin infilling processes, analysing sedimentary histories, and studying geomorphic features. The 1000 m-thick Dasongping section has been well constrained from 7.6–1.8 Ma by magnetostratigraphic (Fig. 1d), which yields SARs varying in a range of 38.2–462.4 m/Ma (Li et al., 2013).

Low-temperature thermochronology is a valuable tool for reconstructing the paleo-topography and the long-term exhumation history of orogenic mountains ranges (Reiners and Brandon, 2006; Herman et al., 2013). 53 thermochronological

data from the eastern and southern margins of DCS were obtained from previously published literatures (Sup Table 2). The numbers of existing thermochronological data per 0.5 Myr intervals were counted, and the frequency vs age of all data from the eastern margin and the southern margin of DCS are shown in the Fig.5, respectively.

## 3 Results

### 3.1 Pollen assemblages

A total of 47 pollen and spore species were identified in 31 samples (Fig.1d) from the Dasongping section. Gymnosperms (with a mean value of 64.1%) and angiosperms (28.0%) are dominant, the content of pteridophytes (including *Polypodium* L., *Pteris* L., *Araiostegia* Cop*., Leucostegia* Presl. *Hicriopteris* Presl. and *Selaginella*) is low (7.9%) and their indication for the environment is not obvious, so pteridophytes are omitted from the spectrum. Conifer trees (mean value of 59.0%) and broadleaf trees (18.4%) are dominant, followed by shrubs and herbs (11.9%) (Fig. 2). The dominant taxa are Pinaceae (20.9%), *Pinus*

(12.5%), *Picea* (11.7%), *Quercus*–evergreen types (*Quercus* E) (10.6%), and *Abies* (6.6%). Pollens of *Tsuga, Keteleeria, Cedrus, Larix, Quercus*–deciduous types (*Quercus* D), *Fagus, Betula, Alnus, Carpinus, Corylus, Ulmus, Liquidambar*, Oleaceae, Poaceae, Asteraceae, and Polygonaceae are common in all analyzed samples. Minor occurrences of Betulaceae, *Juglans, Carya, Tilia, Acer, Ilex,* Apiaceae, Ericaceae, *Melia*, Araliaceae, Fabaceae, Rhamnaceae, Euphorbiaceae, *Nitraria*, and Caprifoliaceae occur in some samples. Variations in the pollen percentages allow the pollen diagram to be divided into

two main zones, Zone I (0–510 m, ~7.6–4.2 Ma) and Zone II (510–1000 m, ~4.2–1.8 Ma), and four subzones (Fig.2).

In Zone I (14 samples, 0–510 m, ~7.6–4.2 Ma), broad-leaved tree pollen (mean of 38.5%), shrubs and herbs (mean of 25.2%) dominate the assemblage, followed by coniferous tree pollen (mean of 17.3%). Broad-leaved tree pollen is a major group, including *Quercus* E (19.0%), *Quercus* D (8.8%), *Alnus* (5.8%), *Ulmus* (3.7%), *Liquidambar* (3.7%), *Corylus* 3.5%), *Carpinus* (3.2%), *Carya* (3.1%), and minor *Fagus, Castanopsis/Castanea, Castanea Mill.*, Betulaceae, *Betula, Juglans, Tilia*

and *Acer*. Herbaceous pollen is also present, including *Compositae* (5.2%), *Oleaceae* (4.3%), *Polygonaceae* (4.3%), and minor *Ilex*, Fabaceae, Euphorbiaceae, and *Melia*. The dominant coniferous taxa include *Pinus* (6.1%), Pinaceae (4.9%), *Picea* (2.2%), *Tsuga* (2.1%), and, with minor *Abies* and *Larix*. Due to unfavorable conditions for pollen conservation, no statistically sufficient pollen and spores were obtained from sandstone and silty sandstone during the period of ~6.3-4.8 Ma. There is more broad-leaved tree pollen in Subzone I-1, whereas there is more coniferous pollen in Subzone I-2 (Fig. 2).

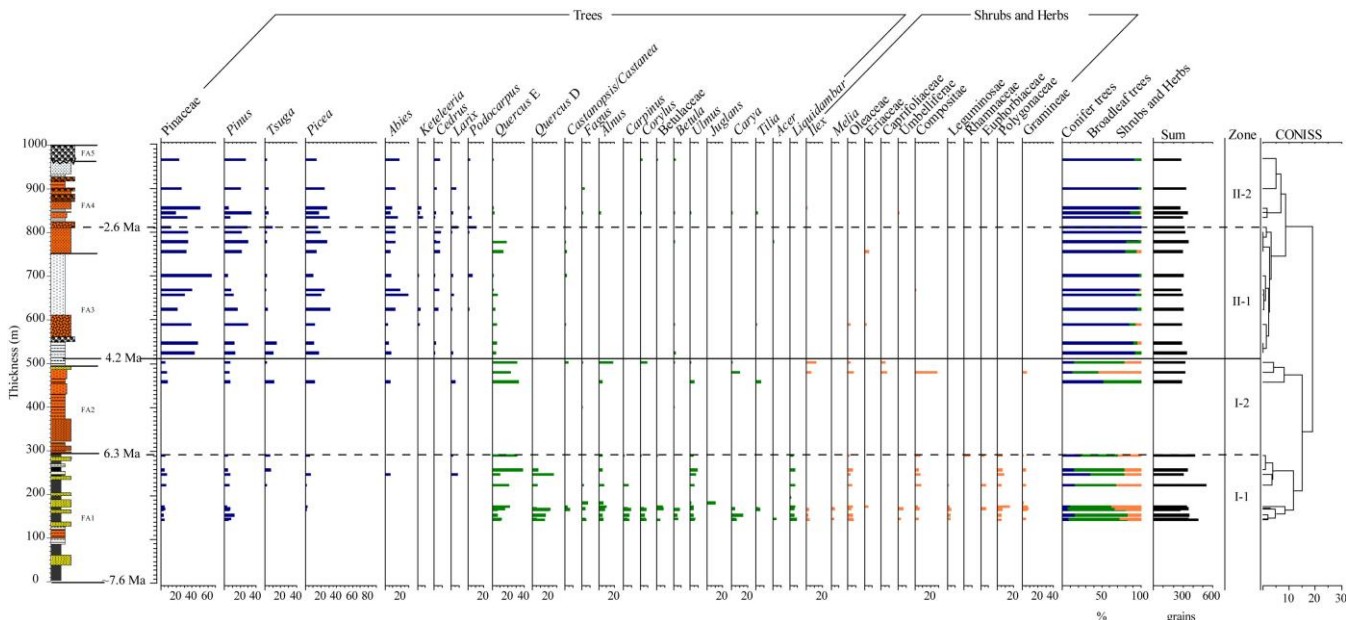

**Figure 2. Variations in lithology, sedimentary facies, and pollen assemblages through the Dasongping section. Lithological captions are same with Fig. 1c.**

In Zone II (17 samples, 510–1000 m, ~4.2–1.8 Ma), coniferous tree pollen (93.3%) dominates the assemblages, followed by minor broad-leaved tree pollen (1.9%) and shrubs and herbs (1.0%). The dominant coniferous taxa include Pinaceae (34.0%), *Picea* (19.5%), *Pinus* (17.9%), *Abies* (11.2%), *Tsuga* (3.4%), *Cedrus* (2.9%), and minor *Keteleeria*, *Larix*, and *Podocarpus*. Broad-leaved tree pollens, including *Quercus* E (3.7%) and minor *Castanopsis/Castanea*, *Fagus*, *Alnus*, *Corylus*, Betulaceae, and *Betula*, are observed in only a few samples. Of note, there are more *Picea*, *Pinus*, *Keteleeria*, *Cedrus*, *Larix*, and *Podocarpus* in Subzone II-2 than in Subzone II-1 (Fig. 2).

Studies of modern four surface samples show that coniferous tree pollens [mainly Pinus (81.3–91.2 %), Tsuga (0.2%–1.1%), and Abies (0%–1.6%)] and fewer broad-leaved tree pollens (mainly Alnus (2.5%–4.9%), Betula (0–3.0 %) and Corylus (0–2.2%)) are present within the elevations of 2400–2600 m a.s.l. from the Dasongping section area. Pollen assemblages are consistent with the modern vegetation distributions, cultivated Pinus forests (2500–3000 m, a.s.l.), on the eastern slope of DCS (Fig. 3).

**3.2 Paleovegetation, *ANNT* and paleoelevation**

From the bottom to top, the modern vegetation belts of the DCS are as follows (Fig.3): (1) Cultivated *Pinus* forests (in the eastern DCS at altitudes of ~2500–3000 m above sea level (a.s.l.); in the western DCS at altitudes of ~1750–2550 m a.s.l.); (2) middle subtropical broadleaf evergreen forest (MTFO; east: absent, west: ~2550–2900 m a.s.l.); (3) cool-temperate mixed forest (COMX; east: ~3000–3350 m a.s.l., west: ~2900–3200 m a.s.l.); (4) *Tsuga dumosa*, broad-leaved forests, and cold

evergreen conifer forest (CLEC; east: ~3350–3800 m asl, west: 3200–3700 m asl); and (5) *Abies delavayi*, tundra with Rhododendron thickets, and Alpine meadow (ALME; east: ~3850–4112 m asl, west: ~3700–4112 m asl). The very flat top of the DCS massif is covered by grasslands, shrubs, and ponds (Fan et al., 2006).

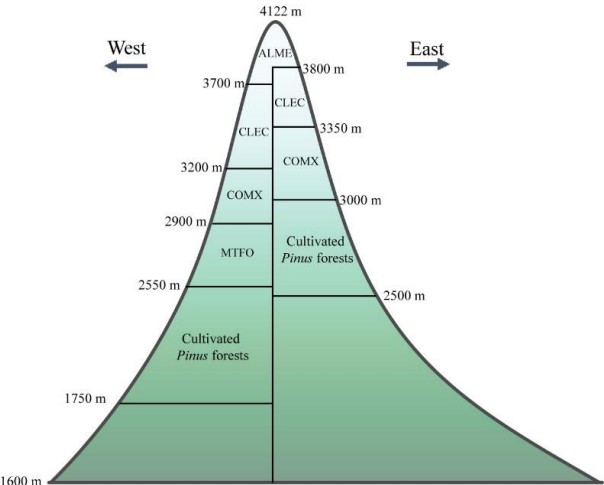

**205** **Figure 3. Vertical distribution of modern vegetation in the Diancang Shang, simulated using the taxa–PFTs–biomes matrixes defined for the China region (Sun et al., 2020). MTFO means middle subtropical broadleaf evergreen forest; COMX means cool-temperate mixed forest; CLEC means cold evergreen conifer forest; ALME means Alpine meadow.**

The dominant vegetation types are MTFO and WAMF (north subtropical mixed forest) during the period of ~7.6–6.3 Ma,
**210** and WAMF at ~4.8–4.2 Ma (Fig. 4a). COMX and CLEC are dominant during ~4.2–1.8 Ma, with the exceptions of WAMF at ~4.2–2.6 Ma (Fig. 4a). To the east of DCS, modern vegetation belts of MTFO, COMX, and CLEC are distributed at altitudes of ~2500–3000, ~3000–3350, and ~3350–3850 m above sea level, respectively (Fig.3). These results indicate that the vegetation in the Dali Basin has been characterized by vertical zonation since the Pliocene. The *ANNT* values in Dasongping section vary in the range of ~3.9–15.1 °C, and have higher mean values of 11.6 °C in Zone I, and lower mean values of 5.3 °C
**215** in Zone II, respectively (Fig. 4b). The mean *ANNT* values of Zone I-1 (~7.6–6.3 Ma), I-2 (~6.3–4.2 Ma), II-1 (~4.2–2.6 Ma) and II-2 (~2.6–1.8 Ma) in the Dasongping section are 11.6±2.3 °C, 12.0±5.1 °C, 6.1±2.0 °C and 4.6±0.5 °C, respectively (Fig.4b).

The *ANNT* of surface samples from 2400 to 2600 m a.s.l and *ANNT* of the late Holecene, mid–Holocene from Lake Erhai coring (elevation of c. 1974 m a.s.l) (Shen et al., 2006) are reconstructed. The mean *ANNT* of surafce samples is 7.9±1.7°C,
**220** which is lower than the modern *ANNT* (~12.0 °C) at the 2500 m a.s.l. The mean *ANNT* of the late Holecene and mid–Holocene from Lake Erhai coring are 11.8±1.3°C and 13.9±1.0 °C, respectivly, both are lower than the *ANNT* (~15.1 °C) at present in the lakeshore (~1970 m a.s.l.) (Shen et al., 2006). However, the *ANNT* in the mid-Holocene from Lake Erhai is ~2.1 °C higher than that in the late Holecene, which is consistent with the fact that the annual mean temperature during the Holocene optimum in south China was generally 2–3 °C higher than it is now (Shen et al., 2006; Zhang et al., 2022). This indicates that the relative

changes in the reconstructed *ANNT* are reliable, even though the reconstructed *ANNT* may possibly be underestimated based on the pollen assemblage of mixed palynological spectra from different altitudes in mountain areas. To mitigate the bias in reconstructing pelaeoelevation using the reconstructed *ANNT* directly, we utilized the temperature mean anomaly, which represents the quantitative differences in *ANNT* values between sedimentary records and surface samples located at 2500 m.a.s.l., to reconstruct the palaeoelevation of DCS. Based on the formula (1) and (2), our reconstruction inferred that the

paleoelevation of DCS may be ~2000–2200 m a.s.l. during ~7.6–4.2 Ma and ~3100–3350 m a.s.l. at ~3.6–1.8 Ma, respectively. The above information implied that the northeastern segment of DCS may have risen by ~1000 m at ~4.2–3.6 Ma.

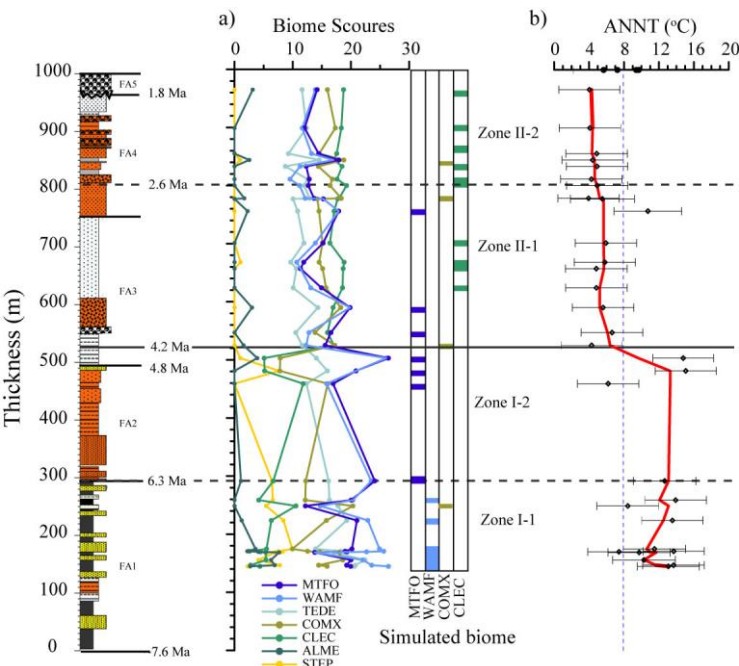

**Figure 4. Variation in lithology, sedimentary facies, a) biome scores and simulated biome type, b) *ANNT* of the Dasongping section.**

## 4 Discussions4.1 Tectonic activities surrounding DCS range during the late Miocene to Pleistocene

The SEMTP may have undergone initial topographic uplift and widespread exhumation from the Eocene to the early Oligocene. The region situated north of 26°N latitude has been at or close to its current elevation since ~40 Ma (Hoke et al., 2014). Additionally, the Mangkang and Gonjo basins, located within the eastern Qiangtang terrane, were likely to have attained elevations of ~2.5-3 km by the end of the Eocene (Su et al., 2019; Tang et al., 2017; Li et al., 2020). The Ailao Shan–Diancang

Shan region underwent ductile deformation that began at least ~34 Ma ago and terminated between 20 and 17 Ma; This deformation was characterized by large-scale differential uplift occurring concurrently with erosion, resulting in the erosion of the mountain range and the adjacent area into a landform with low relief (Tapponnier et al., 1990; Tang et al., 2013; Gourbet et al., 2017;Cao et al., 2021); During the late Cenozoic period, the region experienced dextral ductile-to-brittle transitional

normal faulting marked by regional uplift that was synchronous with the incision of river systems ( Leloup et al., 1993; Leloup et al., 1995; Wang et al., 1998; Wang et al., 2006; Wang et al., 2022). Based on the K-feldspar $^{40}$Ar/$^{39}$Ar ages of dextral sheared mylonites, Leloup et at., (1993) proposed that the present topography of DCS was a consequence of normal faulting along the margin of the mountain, which was initiated at ~4.7 Ma. Geochronological data, including $^{40}$Ar/$^{39}$Ar, K–Ar, and low-temperature thermochronological ages, at different locations in the DCS show that the most recent phase of uplift of DCS ranged from ~5 to ~0 Ma (Cao et al., 2011; Fan et al., 2006; Han et al., 2011; Harrison et al., 1995; Leloup et al., 1993; Li et al., 2012).

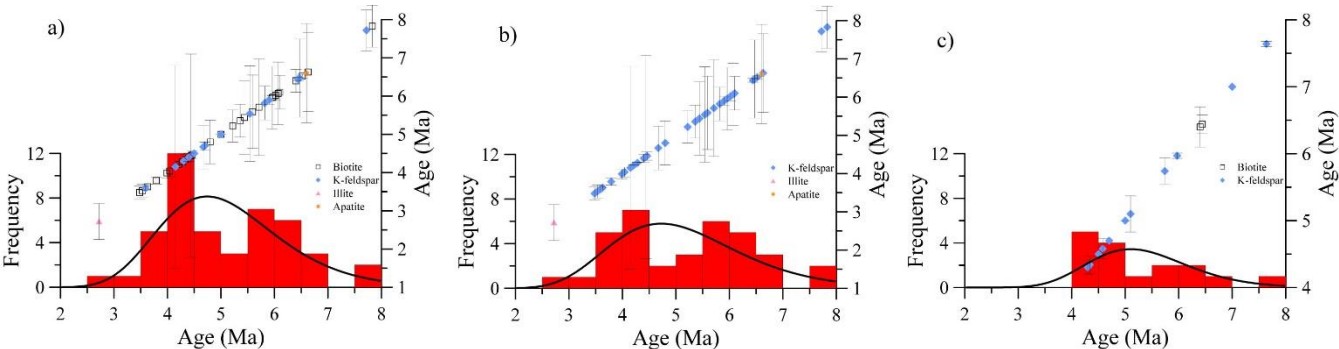

Figure 5. Frequency of thermochronological data from a) the eastern and southern margin of DCS, b) the eastern margin of DCS and c) the southern margin of DCS, respectively (data from references in Table S2).

The synthesis of existing thermochronological data from the eastern and southern margin of DCS cluster between ~7.0 and 3.5 Ma with two high frequency periods at ~6.5–5.5 Ma and ~4.5–4.0 Ma, respectively (Fig.5), which indicated rock cooling events occurred frequently, especially, in the eastern margin of DCS during the period of ~4.5–4.0 Ma. The history of tectonic activities surrounding DCS shear zone can be determined by using thermochronological data and the increase in SAR, the change in provenance, and the timing of conglomerate occurrence in the Neogene sediment sequence. The significant increases in SAR occurred at intervals of ~7.6-6.3 Ma, ~3.6-3.4 Ma and ~2.7-1.8 Ma (Fig.6b), which were corresponding to the initial time of dextral normal faults at ~8 Ma reviewed by geochronological thermochronological data (Wang et al., 2020a), and the intensified normal fault activities at ~3.6 Ma and ~2.7 Ma further indicated by the Late Neogene geochemical sedimentary record in the Dali basin (Zhang et al., 2020). A significant provenance change in infilling of the sedimentary material in Dali Basin occurred at ~4.2 Ma (Fig. 6c-d) (Li et al., 2014; Zhang et al., 2020), which is consistent with the age ~4.5-4.0 Ma of the highest frequency in thermochronological data on time (Fig.5a). The highest mean SAR occurred at ~3.6 Ma with the first occurrence of conglomerates in the Sanying Formation (Figs.1c, 6b), which is consistent with the intensive thermochronological data during ~4.0-3.5 Ma (Figs.5, 6a). Simultaneities between the changes in Neogene sediment sequence and existing thermochronological data from the eastern and southern margin of DCS (Fig.5) indicated that DCS normal fault system was initiated at ~7.6 Ma, intensified at ~4.2 Ma, and enhanced at ~3.6, ~2.7 Ma, respectively. The most recent tectonic

activities in the DCS region have been dominated by differential uplift, which were likely caused by a combination of extension and isostatic adjustments resulting from climate change (Zhang et al., 2001; Wang et al., 2006).

## 4.2 Paleo-vegetation inferences based on palynological assemblages in the Dali Basin

A previous study has shown that pollen grains of *Picea* and *Abies* are large and usually under-represented, whereas pollen grains of *Pinus* are usually over-represented (Erdtman, 1969). Only when its content exceeds 30% can *Pinus* be considered to represent local sources (Li and Yao, 1990; Huang, 1993). As the main components of cold-temperature coniferous forests, *Picea* and *Abies* grow at relatively high altitudes of 3300-4200 m a.s.l. in the mountainous regions of southwestern China, that the percentages of *Picea* and *Abies* predominantly mirror their native habitats, regardless of any potential long-distance migration (Chen et al., 2014; Lu et al., 2008; Miao et al., 2022). *Quercus*–evergreen types encompass alpine oaks, evergreen oaks, and Cyclobalanopsis plants, which are distributed across varying altitudes, resulting in their pollens having distinct climatic indications (Dai et al., 2018; Zhang et al., 2018). For instance, alpine oaks are predominantly found at altitudes ranging from 2500 to 3600 m a.s.l., dominating the high-altitude vegetation communities in the southeast and south regions of the Tibetan Plateau (Zhang et al., 2018; 2020).  Therefore, when analysing and discussing the climatic and ecological significance of *Quercus*–evergreen types pollen, it is necessary to consider the changing characteristics of other pollen types within the pollen assemblage, as well as factors such as the corresponding altitude of the study area.

In the Dasongping section, broad-leaved tree pollen, and herbaceous pollen are dominant in Zone I (0–510 m, ~7.6–4.2 Ma), although small amounts of coniferous tree pollens, including *Picea*, *Tsuga*, and *Pinus*, are also present (Fig. 2). Compared to Zone I, coniferous pollens, including Pinaceae, *Pinus*, *Picea*, and *Abies*, are dominant in Zone II (510–1000 m, ~4.2–1.8 Ma), whereas the percentages of *Tsuga* and broad-leaved trees are substantially lower (Fig. 2). This indicates that coniferous forests have been growing in the surrounding areas with altitudes of at least 3300 m a.s.l. since ~4.2 Ma ago. Notable, *Quercus*–evergreen types (*Quercus* E) are present throughout the Dasongping section, with a mean content of 19.0% in Zone I and 3.7% in Zone II, respectively, showing a significant negative correlation with that of coniferous pollens (Fig.2). This indicates that *Quercus* E in the pollen assemblages is primarily composed of evergreen oak species adapted to low-altitude, relatively warm environments. Otherwise, as the altitude of the region rises and the habitat of alpine oaks expands, their pollen content should also show an increasing trend, similar to that of conifers.

The dominant vegetation types were MTFO and WAMF during ~7.6–6.3 Ma and WAMF at ~4.8–4.2 Ma (Fig. 4a). The inferred vegetation types were similar to those of various adjacent basins. For example, vegetation in the Eryuan Basin comprised mainly mixed forest, suggestive of warm and humid conditions during the period 7.3–4.3 Ma (Wu et al., 2019). In the eastern Eryuan Basin, the main type of vegetation in the Lühe area was mixed forest of coniferous and broad-leaved trees during the latest Miocene (Xu et al., 2008). A mixed coniferous forest has also been inferred from the Sanying Formation in the Lanping Basin and is considered to be of warm-temperate affinity (Huang et al., 2020). COMX and CLEC were dominant during ~4.2–1.8 Ma, with the exceptions of WAMF at ~4.2–2.6 Ma (Fig. 4a). Previous studies have shown that the Eryuan palynoflora during the late Pliocene contained abundant *Pinus* with some *Quercus* and *Alnus* accompanied by herbs such as

*Artemisia* and *Chenopodiaceae*, suggesting that the late Pliocene Eryuan vegetation was vertically zoned and included humid
evergreen broad-leaved forest, needle and broad-leaved mixed evergreen forest, and coniferous forest (Tao and Kong, 1973; Kou et al., 2006; Wu et al., 2019). These results indicate that the vegetation in the Dali Basin has been characterized by vertical zonation since the Pliocene.

## 4.3 Climate change during the late Miocene to Pleistocene in the Dali Basin

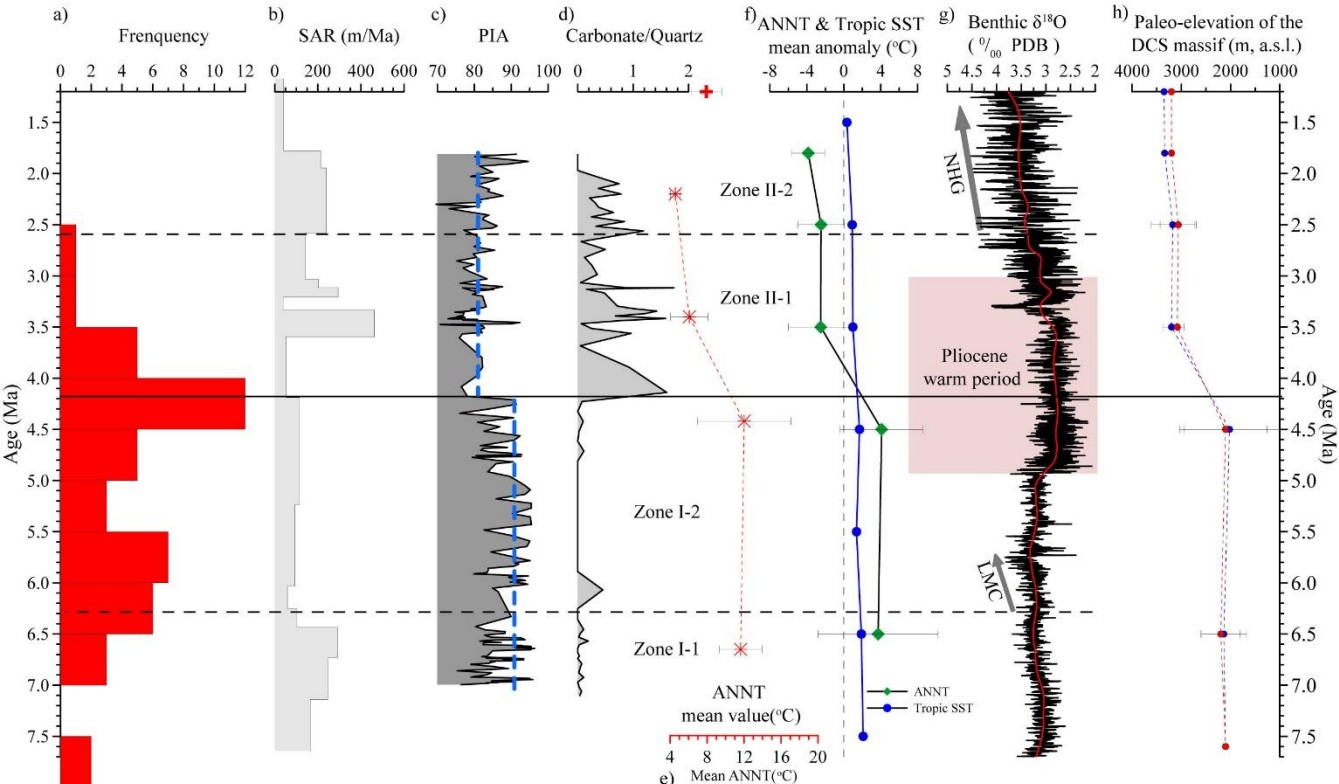

**Figure 6. Comparison of a) Frequency of thermochronological data from the eastern and southern margin of the DCS massif (data from references in Table S1), b) SAR (sediment accumulation rate), c) PIA (plagioclase index of alteration), d) the ratio of carbonate/quartz (from Zhang et al., 2020), e) The mean ANNT value of different zone and f) ANNT mean anomaly per Ma for the Dasongping section and the tropical sea–surface temperature (SST) mean anomaly per Ma since late Miocene (data from Herbert et al., 2016), g) Marine benthic δ¹⁸O stack (Westerhold et al., 2020) on the same timescale and h) Paleoelevation of DCS massif**
**(blue/red solid circle: the common temperature lapse rate of 5/6 °C km⁻¹). LMC: Late Miocene Cooling. NHG: northern hemisphere glaciation.**

The late Miocene was characterized by high chemical weathering intensity and much higher ANNT (Fig. 6c, e). The dominant vegetation types were MTFO and WAMF in Dasongping section (Fig. 4a). A cooling episode during 6.2–5.2 Ma has been
identified from geochemical record in the Zhaotong Basin (Li et al., 2020). However, no ANNT data were achieved during

~6.3–5.0 Ma because of the sandstone lithology in the Dasongping section, whereas plagioclase index of alteration (PIA) showed no particular change (Fig.6c). The tropical SST data (Fig.6f) and global marine $\delta^{18}O$ isotope records (Fig.6g) (Zachos et al., 2001; Westerhold et al., 2020) showed that the late Miocene to late Pliocene climate was relatively warm, except for a reduced zonal SST gradient that prevailed in the tropical Pacific Ocean during the late Miocene (late Miocene Cooling or LMC) (Herbert et al., 2016; Holbourn et al., 2018). Even the values of the ANNT mean anomaly (~3.9 °C) in the Dali Basin were much higher than that of the tropical SST mean anomaly (~1.8 °C), both of them showed very similar variation trends (Fig.6f, g). The generally warm and humid climatic conditions in the Dali Basin during ~7.6–4.2 Ma were almost coinciding with global climatic conditions.

The higher carbonate (calcite and dolomite) contents with less quartz, and much lower values of PIA during the period ~4.2–1.8 Ma (Fig.6c,d) were consistent with the filling by detrital erosion material from bedrock surrounding DCS (Zhang et al., 2020). The *ANNT* showed the maximum decreasing of ~10 °C during ~4.2–3.6 Ma, and then kept at ~4 °C during the period of ~2.6–1.8 Ma (Fig.6b, e). The *ANNT* mean anomaly reduced from 4.1 °C to -2.5 °C in the Dali Basin, and from 1.7 °C to 1.0 °C in the SST mean anomaly (Herbert et al., 2016; Holbourn et al., 2018) (Fig.26), which indicated that the temperature drop in the Dali Basin was significantly higher compared to in tropical SST during ~4.2–3.6 Ma. Noticeable, the Pliocene (~5–3 Ma) was a stable and lasting warm period (Burke et al., 2018). This remarkable *ANNT* reduction of the Dali Basin during the period was in contradiction with the global warm Pliocene climate.

Both *ANNT* anomaly data and tropical SST anomaly data showed a continuous decrease since ~2.7 Ma (Herbert et al., 2016; Holbourn et al., 2018), there had been little difference in the magnitude of temperature mean anomaly decrease between them (Fig.6f). Global climate underwent a major reorganization, marked by a considerable intensification of the Northern Hemisphere Glaciation (NHG) at ~2.75 Ma (Prell and Kutzbach, 1992; Kleiven et al., 2002; Sigman et al., 2004), changing from Pliocene warm climate stage into cold conditions (Zachos et al., 2001; Ravelo et al., 2004; Westerhold et al., 2020). The cold climate changes in the Dali Basin were almost consistent with the variation in global temperature since ~2.7 Ma.

The climatic conditions in the Dali Basin were warm and humid from ~7.6–4.2 Ma, cooler and humid from ~4.2–3.6 Ma, and then cold and humid from ~2.7–1.8 Ma. The remarkable *ANNT* reduction of the Dali Basin during the period of ~4.2–3.6 Ma was in contradiction with the global warm Pliocene climate. Both the provenance changes at ~4.2 Ma and the first occurrence of conglomerates in the Sanying Formation at ~3.6 Ma in the Dali Basin (Figs. 1c, 6b, d) indicated a significant topographic growth of the northern segment of DCS during the period of ~4.2–3.6 Ma (Li et al., 2013; Zhang et al., 2020). The previous climatic reconstruction results for the late Pliocene indicate that the climate conditions in western Yunnan during that period were still slightly warmer and wetter than they are now (Xie et al., 2012; Yao et al., 2012; Su et al., 2013). Given the absence of any pronounced cooling in global surface temperatures (Fig.6g), we proposed that this significant temperature decreasing in the Dali Basin was primarily caused by the rapid topographic uplift of DCS during this period, due to intense tectonic activities, as discussed in Section 4.1.

**5 Constraints on the topography elevation of DCS during the late Miocene to Pleistocene**

Integrating results of tectonic activities surrounding DCS and climatic changes in the Dali Basin, the topography elevation of DCS during the late Miocene to Pleistocene was discussed as follows.

*Stage I (~7.6–5.0 Ma): relatively low DCS*

The paleo-topography of DCS (especially in the northern segment) during ~7.6–5.0 Ma was quantitatively reconstructed as ~2100–2200 m a.s.l. based on the quantitative differences in *ANNT* values (Fig.6f, h). At ~8.0–7.6 Ma the normal fault system initiated and the Dali Basin began to accumulate (Li et al., 2013; Wang et al., 2020a), and the sedimentary environment and provenance were stable during ~7.6–4.2 Ma (Li et al., 2013; Zhang et al., 2020). Thermochronology researches showed that a nearly isothermal phase until to ~5.0 Ma corresponded to a period during which both the geothermal flux remained essentially constant and no significant unroofing occurred in the southern segment of DCS (Leloup et al., 1993; Li et al., 2012). The above information, together with the presence of lake- and/or swamp-facies sediments in the Dasongping section (Fig. 1c), implies that during ~7.6–5.0 Ma, the Dali Basin environment was characterized by a lake and/or swamp surrounded with forest of MTFO/WAMF types at low elevations (<~2200 m a.s.l.) under warm and humid climatic conditions with an absence of significant tectonic activity during this period (Fig.6a).

Therefore, we tentatively conclude that the paleoelevation of DCS likely exceeded 2000 m (may have reached up to ~2200 m) above sea level at ~7.6 Ma, which was consistent with the present average elevation (~2200 m a.s.l.) of the surrounding region in the Dali Basin; Then DCS maintained tectonic stability and relatively warm and humid climatic conditions during ~7.6–5.0 Ma.

*Stage II (~5.0–3.5 Ma): rapid topographic growth of DCS*

The existing thermochronological data were mainly concentrated during the period of ~5.0–3.0 Ma (Fig.5). This period had been promoted as a crucial time for the topographic formation of DCS (Leloup et al., 1993; Wang et al., 2006; Li et al., 2012). The reconstructed paleoelevation of the northern segment of DCS during ~4.2–3.6 Ma changed from ~2100–2200 m a.s.l. to ~3100–3350 m a.s.l. (Fig.6h). Leloup et al. (1993) suggested that the normal fault of DCS initiated at ~4.7 Ma and the southern segment of DCS had been uplifted by at least 4000 m a.s.l. since the deposition began in Erhai Lake. However, the age of older sediments in Erhai Basin is still not precisely known (Guo et al., 1984). As discussed in Stage I, the paleoelevation of the northern segment of DCS likely exceeded 2000 m a.s.l. since the onset of normal faulting of DCS at ~7.6 Ma. Considering average elevation of 2200 m a.s.l. in the surrounding region and 4000 m a.s.l. in the south part of DCS at present, it can be estimated that the southern segment of DCS may have risen by at most ~2000 m rather than 4000 m. Base on the significant growth of ~1000 m between ~4.2 Ma and ~3.6 Ma, we calculated an uplift rate of at least ~1.7 mm/yr. This rate was similar to the uplift rate of ~2 mm/yr estimated by Leloup et al. (1993) and the rate of active normal faulting estimated by Allen et al. (1984).

The existing thermochronological data indicate high-frequency periods at ~5.0–4.0 Ma in the southern segment and at ~4.5–3.5 Ma in the eastern and northern segments of DCS, respectively (Fig.5b, c). These data suggest that the intensified normal fault activities in the southern occurred earlier than those in the northern segment of DCS, which was also consistent

with the roughly northward propagation indicated by the existing chronological data from the SEMTP (Wang et al., 2022). DCS may have risen by at least ~1000 m in the northern and at most ~2000 m in the southern segments. Therefore, we propose that DCS may have reached 3100–3350 m a.s.l. in the north at ~ 3.5 Ma and ~4000 m a.s.l. in the south at ~4.7 Ma, respectively.

Coniferous pollens, including *Pinus*, *Tsuga*, *Picea* and *Abies*, are present throughout the Dasongping section, especially, in Zone II (4.2–1.8 Ma) (Fig.2). These findings indicate that significant vertical climatic zoning of vegetation in DCS may have formed since ~4.7–4.2 Ma, which was consistent with the Pliocene Eryuan vertically zoned vegetation (Tao and Kong, 1973; Kou et al., 2006; Wu et al., 2019).

*Stage III (~3.5–1.8 Ma): intense erosion of DCS*

Taking into account the ongoing global cooling, the paleo-topography of the northern segment of DCS during this period was quantitatively reconstructed as ~3200–3350 m a.s.l. (Fig.6h). As discussed in Stage II, the paleoelevation of the south part of DCS possibly had reached to ~4000 m a.s.l. already. Today, the average elevations of the north and south segments of DCS are 3500 m and 4000 m, respectively. During the period of 3.5–1.8 Ma, only a few low-temperature thermochronological data were available along the eastern margin of DCS, while no data were recorded in the southern margin (Fig.6a, Fig.5), indicating

weak tectonic activity during this time, especially, in the southern segment of DCS. Comparing to the vertical distribution of modern vegetation (Fig. 3), the dominant vegetation of CLEC and COMX during the period of 3.5–1.8 Ma (Fig. 6a) indicates that DCS may have reached its current elevation after ~1.8 Ma, undergoing topographic uplifts of ~150–300 meters in the northern segment of DCS.

    Today, there is a very flat surface on the top of DCS, which can be traced for 100 km from the northern edge to the

405 southern edge of the massif. The existence of a relict surface on the top of DCS suggests that DCS was not subject to unroofing during the late Cenozoic (Fan et al., 2006; Wang et al., 2006). The dominant sedimentation facies were alluvial facie, with a high SAR between ~3.5 and 2.7 Ma, and foothill-pluvial facie became dominant between ~2.7 and 1.8 Ma, with a much higher mean SAR, respectively (Figs.1c, 6a-b). Syntectonic sediments with massive conglomerates developed in the upper ~200 m of the Dasongping section (Fig.1c). These discrepancies between sediment sequence and thermochronological data suggest

that the intensification of SARs were likely caused by intense erosion related to climate cooling. Therefore, we propose that the ongoing global climate cooling, along with the initiation and development of the NHG, played a key role in maintaining the cold and humid climate monsoon conditions in the Dali Basin. This possibly resulted in intense erosion of DCS since 2.7 Ma, ultimately leading to the high SAR in the Dali Basin and the accumulation of ~2200 m thick sediments in the Erhai Lake.

    In conclusion, DCS may have reached a paleoelevation of at least ~2000 m a.s.l. at ~7.6 Ma and maintained tectonic

stability during ~7.6–5.0 Ma. Subsequently, significant uplift of ~1000 m occurred between ~5.0 and ~3.5 Ma in the northern segment of DCS, followed by slow growth during ~3.5–1.8 Ma. Finally, the northern segment of DCS reached an elevation of 3500 m a.s.l. after ~1.8 Ma. Significant changes in vegetation in the Dali Basin since the Late Miocene were primarily due to a marked increase in the paleoelevation of DCS. Ongoing global cooling and associated development of the NHG played a significant role in maintaining cold climatic conditions in the Dali Basin, resulting in intense erosion of DCS since ~2.7 Ma.

Our findings suggest that quantitative paleotemperature reconstruction based on terrestrial pollen assemblages, combined with

geochronological, thermochronological, and sedimentary data, can provide direct constraints on the paleotopographic evolution of mountains.

**Data availability.**

The research data used in this study are available via Science Data Bank (https://www.scidb.cn/anonymous/VWZRbjZm).

**Author contributions.**

CXZ and HBW designed the study, CXZ carried out data preparation. XLZ and YKD conducted pollen analyses. YXJ and WCZ conducted climate reconstruction analyses. CXZ generated the figures. CXZ, HBW, SHL and CLD contributed to the writing of the article.

**Competing interests.**

The contact author has declared that neither of the authors has any competing interests.

**Acknowledgments**

We thank Dr. Zhilin He for assisting us with Figure 1a. We are grateful to two anonymous reviewers and Prof. Tao Su for their constructive reviews and meaningful comments. This study was financially supported by the National Natural Science Foundation of China (Grants No. 42072209; 42488201).

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
