# Peer review of "Rapid topographic growth of the Diancang Shan, southeastern margin of the Tibetan Plateau since 5.0–3.5 Ma"

_Climate of the Past, 2024_

## Author Comment (AC1)

Supporting Information for

**Rapid topographic growth of the Diancang Shan, southeastern margin of the Tibetan Plateau since 5.0–3.5 Ma**

**Chunxia Zhang[1,2], Haibin Wu[1,3], Xiuli Zhao[2], Yunkai Deng[1,3], Yunxia Jia[4], Wenchao Zhang[1], Shihu Li[5], Chenglong Deng[3,5]**

[1]Key Laboratory of Cenozoic Geology and Environment, Institute of Geology and Geophysics, Chinese Academy of Sciences, Beijing 100029, China

[2]College of Earth Science and Engineering, Shandong University of Science and Technology, Qingdao 266590, Shandong Province, China

[3]University of Chinese Academy of Sciences, Beijing, China

[4]School of Geographical Science, Shanxi Normal University, Taiyuan 030031, China

[5]State Key Laboratory of Lithospheric Evolution, Institute of Geology and Geophysics, Chinese Academy of Sciences, Beijing 100029, China

*Correspondence to*: Chunxia Zhang (cxzhang@mail.iggcas.ac.cn)

**Contents of this file**

Text S1

Figure S1

Table S1, S2

**Introduction**

This supplementary material contains one section of text (Text S1), one supplementary figure (Figure S1), and two tables (Table S1, S2).

**Text S1**

**Climate reconstruction method base on pollen data**

The Modern Analogue Technique (MAT) is a commonly used method for reconstructing past climate (Overpeck et al., 1985). This approach was based on the measure of the degree of similarity between fossil pollen and modern pollen (Chevalier et al., 2020).

This analogue-based approach avoided to fit the pollen-climate models, thus outperformed other regression-based approach (such as weighted averaging-partial least squares, WAPLS) (ter Braak and Juggins, 1993; Zhang et al., 2022). However, this approach might suffer from the so-called 'no-analogue' problem (Chevalier et al., 2020). The pollen taxa-PFT transformation scheme was later developed to address this problem (Peyron et al., 1998). The plant function types (PFTs) are groups of plants characterized by common phenological and climate constraints, thus the taxa with no modern analog can be replaced by other taxa within the same PFT group (Mauri et al., 2015; Prentice et al., 1992). The PFT-based MAT method has been widely employed in the paleoclimate reconstruction (Davis et al., 2003; Peyron et al., 1998; Zhang et al., 2022).

[Figure]

**Figure S1.** Scatter plots and residuals of observed vs. predicted *ANNT* based on PFT-based Modern Analogue Technique (MAT, k=6) and weighted-average partial least squares (WAPLS, component=3) estimated by bootstrapping cross-validation.

**Table S1.** The thickness, lithology of samples for pollen analyses, and the total sum of pollen grains for each sample.

[revised manuscript text omitted]

---

## Author Response (AR1)

**cp-2024-43 Title: Rapid topographic growth of the Diancang Shan, southeastern margin of the Tibetan Plateau since 5.0–3.5 Ma**

Dear editor,

I enclose a revised version of our manuscript "Rapid topographic growth of the Diancang Shan, southeastern margin of the Tibetan Plateau since 5.0–3.5 Ma" (cp-2024-43), in which all the points raised by the two reviewers and one community have been taken into consideration.

The main revisions we have made are marked in revised manuscript. In particular these are:

(1) The methodology for reconstructing elevation has been moved to the Methods section as "2.2.3 Paleoelevation reconstruction". The results of elevation reconstruction have been presented in Results section under "3.2 Paleovegetation, *ANNT* and paleoelevation".

(2) Contents regarding tectonic activities in the southeastern margin of the Tibetan Plateau and the possible mechanism for the topographic growth history of the studied region have been included in Section 4.1.

(3) A new Section 4.2, titled 'Paleo-vegetation Inferences Based on Palynological Assemblages in the Dali Basin,' has been included in the revised manuscript.

(4) The original section 4.3 and section 5 have been restructured as Section "5 Constraints on the topography elevation of DCS during the late Miocene to Pleistocene".

We believe that changes in the revised manuscript incorporate all substantial suggestions of comments from the reviewers wherever it is feasible.

Yours sincerely,

Chunxia Zhang

E-mail: cxzhang@mail.iggcas.ac.cn

**Replies to reviewers' comments point to point**

.........................................................

Reply to Reviewer #1:

.........................................................

We thank the reviewer for his/her helpful comments to the manuscript. In the following we provide a point-by-point response to the reviewer's comments.

The manuscript "Magnetostratigraphy of the Dali Basin in Yunnan and implications for late Neogene rotation of the southeast margin of the Tibetan Plateau" provided by Zhang and co-authors is well designed, and the result seems very interesting. I agree to accept it after moderate modification.

**Response:** We thank you for the overall positive evaluation of our work. In the following we provide point-by-point response to their comments.

The following comments are provided for reference:

**1**: Some key names in the text, such as the Ailao Shan-Red River shear zone, Eryuan Basin, and Erhai Basin etc., should be shown in Figure 1. Additionally, the font color in Figure 1b, being white, is not sufficiently legible. It is advisable to opt for a more distinct color for better visibility. Moreover, it is suggested to enhance Figure 1 by including a map displaying the altitude distribution of the four surface samples collected from elevations ranging between approximately 2400 and 2600 meters in the Dasongping section area.

**Response:** Thanks for your valuable comments. Figure 1 has been revised based on your comments. Some key names, including of the Ailao Shan-Red River shear zone, Eryuan Basin, and Erhai Basin etc. have been added in Figure 1a and 1b. The font color in Figure 1b has been adjusted. A new small map has been added in Figure 1c to display the altitude distribution of the four surface samples. Please see the details in the revised Figure 1.

[Figure]

**Figure 1. (a) Location of the study area. (b) Shaded relief map of the Dali Highlands and the Dali fault system, as well as the Diancang Shan. The distribution and configuration of active faults are modified after Wang et al. (1998). (c) The altitude distribution of the four surface samples. (d) Lithologies, sedimentary facies, and samples for pollen, bulk mineral, and geochemical analyses. DCS: Diancang Shan. Solid blue circles in (d) correspond to samples that yielded abundant pollen grains.**

**2:** The scatter plot of the PFT- based MAT method (a) in Figure S1 exhibits a higher dispersion compared to the scatter plot of the WAPLS method. There are a greater number of extreme values with deviations (b) ≥10 or ≤-10 in the PFT- based MAT method, making it challenging to clearly identify the superiority of the PFT- based MAT method over the WAPLS method from the graph. Please refer to the figure below for illustration.

[Figure]

**Response:** Thanks for the insightful comments. The dispersion of PFT-based MAT results is indeed higher than that of the WAPLS method. This phenomenon is also observed in other model evaluations in the North China (Liu et al., 2020).

The MAT method is based on the similarity between fossil and modern taxa (Overpeck et al., 1985). As the modern taxa may be impacted by non-climatic factors (human activity, altitude, etc.), the reconstructions might be dispersed. The WAPLS method assumes a unimodal taxon-climate response model (ter Braak et al., 1993), thus the deviations might be smoothed. We have performed the leave-one-out (LOO) and h-block (HB) cross-validation to evaluate the method performance (Zhang et al., 2022a,b). For the annual mean temperature in the East Asia, the performance of MAT methods ($R^2$=0.80 for LOO, $R^2$=0.62 for HB) is better than the WAPLS method ($R^2$=0.69 in LOO, $R^2$=0.64 in HB) (Zhang et al., 2022a). It is also the case for other climate variables (annual mean precipitation, temperature of the warmest (MTWA) and coldest months (MTCO)) (Zhang et al., 2022a, b). We added sentences in the revised manuscript, "It is reasonable that the scatter plots of PFT-based method exhibits higher dispersion than those of the WAPLS method, because the modern analogues might be impacted by non-climatic factors (Zhang et al., 2022b)".

**References:**

Liu L, Wang W, Chen D, et al. Soil-surface pollen assemblages and quantitative relationships with vegetation and climate from the Inner Mongolian Plateau and adjacent mountain areas of northern China. Palaeogeography, Palaeoclimatology, Palaeoecology. 2020;543:109600.

Overpeck JT, Webb T III, Prentice IC. Quantitative Interpretation of Fossil Pollen Spectra: Dissimilarity Coefficients and the Method of Modern Analogs. Quaternary Research. 1985;23(1):87-108.

ter Braak CJF, Juggins S. Weighted averaging partial least squares regression (WA-PLS): an improved method for reconstructing environmental variables from species assemblages. Hydrobiologia. 1993;269-270(1):485-502. doi:10.1007/bf00028046

Zhang W, Wu H, Cheng J, et al. Holocene seasonal temperature evolution and spatial variability over the Northern Hemisphere landmass. Nature Communications. 2022;13(1).

Zhang W, Wu H, Li Q, Liu Z, Cheng J. Large training dataset is crucial for analogue-based precipitation reconstruction during the early Holocene. Science Bulletin. 2022b;67(11):1118-1121.

3:I suggest unifying the word of "WAPLS" throughout the manuscript, for example, "WA-PLS" on page 124.

**Response:** We have unified all instances of "WA-PLS" to "WAPLS" in the revised manuscript based on your suggestion.

4:The manuscript notes that due to unfavorable conditions for pollen preservation, insufficient pollen and spores were gathered from sandstone and silty sandstone between approximately 6.3-4.8 million years ago. To improve clarity, it is advisable to provide a specific statistic indicating failure to meet the criteria, such as none reaching a minimum of 100 pollen grains. Moreover, if the count surpasses 100 pollen grains for these samples, it is advised to incorporate this detail as supplementary material for reference figures.

**Response:** Thank you for your suggestion. In fact, only less than 5 pollen grains or none were gathered from sandstone, silty sandstone and red siltstone samples. Nevertheless, the thickness, lithology of the samples for pollen analyses, and the total sum of pollen grains for each sample have been provided in Sup Table 1. Please see the details in the revised manuscript and the Sup Table 1.

5:The evolution of paleotopography is crucial in the study. I suggest incorporating the methodology for reconstructing elevation into the Methods section, presenting the results of elevation reconstruction in the Results section, and concluding with a discussion on the topographic evolution.

**Response:** Thank you for your valuable suggestions, we accepted them all. The methodology for reconstructing elevation has been moved to the Methods section as "2.2.3 Paleoelevation reconstruction". The results of elevation reconstruction have been presented in Results section under "3.2 Paleovegetation, ANNT and paleoelevation". The original section 4.3 and section 5 have been restructured as Section "5 Constraints on the topography elevation of DCS during the late Miocene to Pleistocene". Please find the details in the revised manuscript.

Yours sincerely,

Chunxia Zhang

..........................................................

..............................................
Reply to Reviewer #2:

..............................................
First of all, we wish to acknowledge the reviewer for his/her helpful comments to the manuscript. In the following we provide point-by-point response to their comments.

In this study, the authors quantitatively reconstructed changes in annual mean temperature based on the stratigraphic record of palynological proxies from a well-dated late Miocene-Pleistocene sequence in the Dali Basin, southeastern Tibetan Plateau. They integrated these data with thermochronological data from the eastern and southern margins of Diancang Shan to clarify the topographic evolution of Diancang Shan since the late Cenozoic. After excluding the influence of global temperature changes, the topographic evolutions of Diancang Shan were estimated. The results provide direct constraints on the paleotopographic evolution of mountains. This study has solid field investigations, sampling, and laboratory experiments, with clear thinking, appropriate methods, and reliable results,which provided new theoretical basis for global change research. It is suitable for CP. There are several issues to consider when improving.

**Response:** We are grateful for your overall favorable assessment of our work, and we would like to address each of your comments in detail as follows.

1. In Yunnan, evergreen oak may mainly be alpine oak. The alpine oak is different from the evergreen oak in eastern China, as it grows at a relatively high altitude. This type of change in the diagram is very interesting, and it is recommended to supplement the ecological analysis of alpine oak in the results and discussion.

   **Response:** We have incorporated your comment and have added a new Section 4.2, titled 'Paleo-vegetation Inferences Based on Palynological Assemblages in the Dali Basin,' to the revised manuscript. Please refer to the revised manuscript for detailed discussions on the *Quercus* – evergreen types.

2. The transition of climate environment from Zone 1 to Zone 2 does not match the results of oxygen isotopes, which can increase the discussion of some mechanisms.

   **Response:** Thank you for your comment. More discussions have been included in Section 4.3 as follows: The climatic conditions in the Dali Basin were warm and humid from ~7.6－4.2 Ma, cooler and humid from ~4.2－3.6 Ma, and then cold and humid from ~2.7－1.8 Ma. The previous climatic reconstruction results for the late Pliocene indicate that the climate conditions in western Yunnan during that period were still slightly warmer and wetter than they are now (Xie et al., 2012; Yao et al., 2012; Su et al., 2013). Given the absence of any pronounced cooling in global surface temperatures (Fig.6g), we proposed that this significant temperature decreasing in the Dali Basin was primarily caused by the rapid topographic uplift of DCS during this period, due to intense tectonic activities, as discussed in Section 4.1.

3. How to test the quantitative results of ancient climate? Insufficient comparative analysis.

**Response:** Thank you for your comment. There are relatively few quantitatively reconstructed paleoclimate records with precise chronological age control in the southwestern China region since the Late Miocene, which are insufficient for comparative analysis. However, our quantitative reconstruction method provides a high degree of reliability for paleotemperature reconstruction. In our work, we used the Plant Functional Types (PFTs)-based Modern Analogue Technique (MAT) method to quantitatively reconstruct the Annual Mean Temperature (ANNT). Plant Functional Types (PFTs) are groups of plants characterized by common phenological and climate constraints. The predicted ANNT based on the PFTs-based MAT showed a more significant linear relationship ($R^2$ =0.83) with observed ANNT compared to the Weighted-Average Partial Least Squares (WAPLS, component=3) method ($R^2$ =0.80), as estimated by bootstrapping cross-validation. As the most effective climate reconstruction method based on pollen data, the PFT-based MAT method has been widely employed in paleoclimate reconstruction. More information has been included in the revised manuscript.

4. Discussions on research connections with other regions of the Qinghai Tibet Plateau can be added.

**Response:** Thank you for your comments. Contents regarding tectonic activities in the southeastern margin of the Tibetan Plateau and the possible mechanism for the topographic growth history of the studied region have been included in discussions Section 4.1.

Yours sincerely,

Chunxia Zhang

.............................................................

.............................................................

Reply to community:

.............................................................

First of all, we wish to acknowledge Prof. Su's helpful comments to the manuscript. In the following we provide point-by-point response to his comments.

The history for the growth of the Tibetan Plateau is still largely unresolved because evidence with reliable proxies is far from enough. The authors reconstructed the topography history of southeastern margin of the Tibetan Plateau since the late Miocene using well-dated palynological and thermochronological data. Even there have been many studies concerning on this topic in the region, most of them are based on individual evidence from geochemistry, thermochronology, or macrofossils. Palynological investigations are limited and not in long time interval. Therefore, this study would undoubtedly improve the understanding for the growth history of southeastern margin of the Tibetan Plateau. I am satisfied with most contents, and have several major suggestions as below.

**Response:** We appreciate your overall positive evaluation of our work. In the following, we provide

a point-by-point response to your comments.

My major concern is the palaeoelevation reconstruction with palynological investigation. The result indicates that the mean ANNTs derived from surface samples and Lake Erhai coring are lower than the modern ANNTs at the similar elevations, which might be due to the mixed palynological spectra from different altitudes as pointed out by the authors; therefore, a cooler climate is expected. If this is the case, there should be bias for palaeoelevation reconstruction using pollen and spore data. I am wondering if the authors could consider the factor of transportation with the percentages of pollen/spores of each taxon.

**Response:** Thank you for your comment. We try to explain the reliability of our palaeoelevation reconstruction from two aspects. Firstly, we used the PFTs-based modern analogue technique (MAT) method to quantitively reconstruct the annual mean temperature (ANNT). Plant function types (PFTs) are groups of plants characterized by common phenological and climate constraints, not pollen species. Therefore, the taxa with no modern analog can be replaced by other taxa within the same PFT group, thus avoiding the "no-analogue" problem. Therefore, as the best climate reconstruction method base on pollen data, the PFT-based MAT method has been widely employed in the paleoclimate reconstruction. Secondly, our results show that, despite the potential for underestimation of reconstructed ANNTs based on mixed palynological spectra from varying altitudes in mountain regions, the relative changes in the reconstructed ANNTs are reliable. For instance, the ANNT in the mid-Holocene from Lake Erhai is ~2.1 $^{\circ}$C higher than that in the late Holecene, which aligns with the fact that the annual mean temperature during the Holocene optimum in south China was generally 2–3 $^{\circ}$C higher than present-day temperatures (Shen et al., 2006; Zhang et al., 2022). In our manuscript, to mitigate the bias in reconstructing pelaeoelevation using the reconstructed ANNT directly, we utilized the temperature mean anomaly, which represents the quantitative differences in ANNT values between sedimentary records and surface samples located at 2500 m.a.s.l., to reconstruct the palaeoelevation of DCS. Above information has been included in revised manuscript.

I suggest reorganizing Discussion to better show the significance of this study. Most contents in Discussion focus on the data derived from Dali Basin, e.g., in part 4.1, there are thermochronological data with different geological ages along DCS without comparison of similar studies in the adjacent areas of southeastern margin of the Tibetan Plateau. In recent years, there have been increasing studies in basins near Dali Basin, which indicate that the modern-like topography might have been established by the late Eocene-early Oligocene, e.g., Baoshan, Jiangchuan, Mangkang, Relu, and Gongjue basins. Only the spatial/temporal comparison from different basins was applied could we better understand the topographic history of southeastern margin of the Tibetan Plateau. Besides, the authors may briefly explain for the possible mechanism for the topographic growth history of the studied region.

**Response:** We have accepted your comments, and part 4.1 has been reorganized in the revised

manuscript. Contents regarding tectonic activities in the southeastern margin of the Tibetan Plateau and the possible mechanism for the topographic growth history of the studied region have been included in part 4.1.

As mentioned by the authors that both pollen grains and spores were observed in the palynological investigation of the Dasongping section; however, I could not find the spore data in the main text and Figure 2 of part 3. Was spore percentages were low or ignored?

**Response:** We have accepted your comments. One sentence "Gymnosperms (with a mean value of 64.1%) and angiosperms (28.0%) are dominant, while the content of pteridophytes (including *Polypodium* L., *Pteris* L., *Araiostegia* Cop*., Leucostegia* Presl. *Hicriopteris* Presl. and *Selaginella*) is low (7.9%) and their indication for the environment is not obvious, so pteridophytes are omitted from the spectrum." has been included in part 3.

Yours sincerely,

Chunxia Zhang

............................................................